# Tubeless Ureterorenoscopy-Our Experience Using a 120 W Laser and Dusting Technique: Postoperative Pain, Complications, and Readmissions

**DOI:** 10.3390/jpm12111878

**Published:** 2022-11-09

**Authors:** Guy Verhovsky, Yishai H. Rappaport, Dorit E. Zilberman, Amos Neheman, Amnon Zisman, Ilan Gielchinsky, Leon Chertin, Itay M. Sabler

**Affiliations:** 1Department of Urology, Shamir Medical Center, Assaf Harofeh Campus, the Sackler School of Medicine, Tel Aviv University, Tel Aviv 6997801, Israel; 2Department of Urology, Shamir Medical Center, Assaf-Harofeh Zeriffin 7030000, Israel; 3Department of Urology, Samson Assuta Medical Center, Ashdod, the Goldman School of Medicine, Ben-Gurion University of the Negev, Beer Sheva 8410501, Israel; 4Department of Urology, Sheba Medical Center, the Sackler School of Medicine, Tel Aviv University, Tel Aviv 6997801, Israel

**Keywords:** ureteroscopy, tubeless, nephrolithiasis, stent, pain

## Abstract

Introduction and Objective: Both double J-stent (DJS) and ureter catheter (UC) drainage represent routine practice following ureterorenoscopy. In select situations, a tubeless approach is possible and safe. In tubeless cases, we use a sheathless dusting technique with the Lumenis^®^ MOSES Pulse™120 H Holmium: YAG laser. We evaluated these three drainage subgroups and compared postoperative pain, complications, and readmissions. Methods: A retrospective database of 269 consecutive patients who underwent primary ureterorenoscopy for the treatment of upper urinary tract stones between October 2018 and August 2019. The cohort was divided according to post-operative drainage as Tubeless, UC, and DJS. The decision on whether to perform post-operative drainage was by surgeon preference. Demographic and clinical parameters such as stone location, number, and burden, hydronephrosis grade, and postoperative complications (fever, acute renal failure, and the obstruction of the upper urinary tract by Stone Street) were assessed. Pain was assessed using a 0–10 Visual Analog Scale score (VAS) and the use of analgesics by dose/case in each group. Results: There were 70 (26%) tubeless, 136 (50%) UC, and 63 (24%) DJS cases. Patients drained with DJSs had a significantly higher stone burden, more severe obstruction, and prolonged operative time. Tubeless and UC-drained patients had the same stone characteristics with maximal diameters of 8.4 (6.1–12) mm and 8 (5.2–11.5) mm in comparison to the stented group, with 12 (8.6–16.6) mm, *p* < 0.01. The operation time was the longest in the stented group at 49 min (IQR 33–60) in comparison to the UC and tubeless groups at 32 min (23–45) and 28 min (20–40), respectively (*p* < 0.001). Auxiliary procedures were more prevalent in the stented group, but the overall stone-free rate was not significantly different, *p* = 0.285. Postoperative ER visits, readmissions, and complications were the highest in the UC-drained group, at 20% in the UC vs. 6% in the tubeless and 10% in the stented groups. Post-operative pain levels and analgesic use were significantly lower in the tubeless group with a significant reduction in opiate usage. Conclusions: A tubeless approach is safe in selected cases with fewer post-operative complications. While DJS should be considered in complex cases, UC may be omitted in straightforward cases since it does not appear to reduce immediate postoperative complications. Those fitted for tubeless procedures had improved postoperative outcomes, facilitating outpatient approach to upper urinary tract stone treatment and patient satisfaction.

## 1. Introduction

Post-ureterorenoscopy drainage is a common routine practice among endourologists and is thought to prevent postoperative complications and enhance stone fragment expulsion, as well as reduce the probability of ureteral obstruction due to blood clots and stone fragments [1,2,3,4,5].

In recent years, the development of thinner flexible and semi-rigid ureterorenoscopes with improved optics and advancements in laser energy delivery has advanced the field of endourology considerably. These advancements have enabled reduced indwelling time for ureteral catheters connected to a urethral catheter and they are suitable substitutes for DJSs. Such drainage prevents post-operational ureteral swelling or edema and avoids stent-related complications [5,6].

Recently, several publications assessed the feasibility of a tubeless approach, especially in non-complicated upper urinary tract stone treatment. Densted and colleagues concluded that in uncomplicated ureteroscopies without ureteral dilatation, patients left tubeless had significantly fewer postoperative stent-related symptoms with no difference in long- or short-term complications [7]. 

In contrast, a meta-analysis by Wang et al. appraised data from 22 randomized controlled trials, concluding that DJSs were valuable in preventing re-hospitalizations, with the caveat of increased complications without improved stone-free rates [8].

In this study, we describe our current endourological drainage practice routine, following a sheathless dusting procedure, using a high-power Lumenis^®^ MOSES Pulse™ 120 H Holmium: YAG laser. We evaluated and compared DJS and UC as valid drainage options and compared them to tubeless ureterorenoscopy-selected cases. 

## 2. Methods

After Institutional board review approval, 269 consecutive patients who underwent primary ureterorenoscopy for the treatment of upper urinary tract stones between October 2018 and August 2019 were evaluated. The retrospective cohort was divided according to post-operative drainage as Tubeless, UC, and DJS. All patients were operated on by three experienced senior endourology surgeons and the choice of whether to leave drainage was by surgeon preference. Larger stones required the use of a ureteric access sheath, and more complex cases were drained by DJS and indwelled for up to two weeks. Cases requiring complementary procedures were drained by DJS as well. In uneventful cases, two surgeons left UC as their routine and the third completed the procedures as tubeless. Demographic and clinical parameters such as stone location, number, and burden, hydronephrosis grade, and postoperative complications (fever, acute renal failure, and stone street obstructing the upper urinary tract) were analyzed.

Ureteral and renal stones were treated using 6.5/8.5 F and 8.9/9.8 F Richard Wolf^®^ semi-rigid Dual-channel ureteroscopes or a Storz^®^ 7.5 F flex X^2^ ureteroscope. For dusting, we used 265 μm or 365 μm laser fibers. The laser lithotripsy was performed using a Holmium: Yag laser (PulseP 120 H, Lumenis Inc., Carmiel, Israel); the following laser settings were used (dusting in situ 0.6–0.8 Joule, 12–15 Hz, fragmentation 0.8–1 Joule, 8–10 Hz; Pop dusting 0.6 Joule 50 Hz). When a semirigid ureteroscope was used, the stones were fragmented and pulled out using a 2.2 F Cook^®^ N-circle Nitinol stone extractor, when possible, whereas, when the flexible ureteroscope was used, a sheathless dusting technique was applied. The procedure was terminated when there were no apparent stone fragments left in the ureter or complete renal stone dusting was achieved. In patients with a high stone burden, severe preoperative hydronephrosis, impacted ureteral stones, previous infection, or when there was doubt of incomplete fragmentation due to suboptimal visual conditions, a DJS was left at the end of the procedure, and auxiliary intervention was scheduled. Patients left tubeless were those with relatively small stone volumes, patients pre-stented due to acceptable emergency indications, patients without previous chronic obstruction or in cases devoid of intraoperative complications, such as ureteric trauma or perforation, and in those with evidence of efficient dusting without apparent stone fragments larger than 2–3 mm. In borderline cases, the temporary drainage of the treated upper urinary tract was performed. Either a 5 F UC attached to a urethral catheter was utilized for 24 h or a DJS was left. The DJS was retained for 7–10 days or until a supplementary procedure. Patients were discharged at postoperative day (POD) 1 or POD 2, depending on their postoperative course.

Postoperative analgesia protocol included Diclofenac 75 mg IM, Dipyrone 1 g, and Tramadol 100 mg IV combined with 10 mg Pramin and 10 mg Morphine (MO) in 100 CC normal saline, depending on pain severity. The pain was assessed by the Visual Analog Scale (VAS) and the use of analgesics by dose/case in each group. 

### Statistical Analysis

We used continuous variables to describe the median (interquartile range [IQR]) and frequency (proportions) to describe categorical variables. The categorical dependent variables were evaluated using chi-square or Spearman’s correlation tests, whereas continuous dependent variables were analyzed using Pearson correlation. The Wilcoxon signed-rank test was used to compare continuous variables. A Kruskal–Wallis test was performed in comparison between abnormally distributed groups. Statistical significance was defined as *p* < 0.05, and all analyses were performed using SPSS software version 23 (SPSS Inc., Chicago, IL, USA). 

## 3. Results

Out of 269 consecutive uretrorenoscopies, 136 (50%) were drained with UC, 63 (24%) with DJS, and 70 (26%) were left tubeless. The maximal diameter of the largest stone in the UC and tubeless groups were similar and smaller than in the DJS group 8.4 mm (6.1–12), 8 mm (5.2–11.5), and 12 mm (8.6–16.6), respectively, *p* < 0.0001. The operation times were 32 (23–45) and 28 (20–40) minutes in the UC and tubeless groups, respectively, and 49 (33–60) minutes in the stent group, *p* < 0.0001. Tubeless patients were mostly pre-stented, 58% vs. 41% in the UC group and 8% in the stented, *p* < 0.01. No significant difference in hospitalization time between the groups was observed. Auxiliary procedures were significantly more prevalent in the stented and UC groups; however, the overall stone-free rate did not differ: 102/124 (82%) and 58/61 (95%) in the UC and tubeless groups, and 54/56 (96%) in the stent group, *p* = 0.285 (Table 1). Postoperative pain scores were significantly lower in the tubeless group (VAS 1.75 ± 2.79) in comparison to the stented (2.82 ± 2.56) and the UC-drained (2.87 ± 2.78), *p* = 0.03 (Table 2). In the UC group, the patients required significantly more dipyrone, tramadol, diclofenac, and morphine. No patient in the tubeless group required morphine (Table 3).

## 4. Discussion

Post-ureterorenoscopy stenting is still a common endourological practice [1,9,10], despite both the European and AUA guidelines currently not recommending routine DJ-stent insertion following URS [11]. Nonetheless, DJ-stent insertion was carried out in 86.2% of patients undergoing laser lithotripsy and in 70.5% of patients undergoing basket retrieval [9].

Our data of 269 consecutive ureterorenoscopies demonstrate the common reluctance for the tubeless approach; 74% of patients were left with short-term UC or DJ-stents. Comparing the three groups of patients, those tubeless and those drained with UC had similar stone volumes, stone locations, pre-stenting rates, and operation times. DJ-stented patients had larger stones, mostly located in the kidney, longer operation times, and more supplementary procedures. Interestingly, patients drained with UC had more ER visits and a higher type 1 Clavien-Dindo complication rate. 

Technological evolution in the past several decades has enabled smaller diameter flexible ureterorenoscopes and laser machines that enable more efficient stone fragmentation while reducing the complication potential [12]. The transition to a 120 W high-power Ho:YAG system (Pulse 120 H; Lumenis, Israel, Carmiel Inc.) enabled dusting pulse frequencies of 40–80 Hz and energy settings ranging from 0.2 to 0.6 J.

Accumulating evidence comparing the routine post-URS DJ-stenting and tubeless procedure did not find significant differences in SFR, the incidence of UTI, ureteral strictures, or unplanned medical visits [5,6,7,8,13,14]. Tang et al., in their Cochrane analysis, also found lower incidences of dysuria, urinary frequency, and hematuria in tubeless simple URS. Reduction in postoperative pain was equivocal [15]. Ordonez et al. published a Cochrane analysis of 23 trials evaluating 2656 patients who were treated with or without a stent after primary URS, concluding that they “failed to demonstrate the merits of routine postoperative stent,” as no improvement in reintervention rates, rehospitalization, or stricture formation was found [1].

However, the authors of both Cochrane analyses criticized the quality of the methodology and concluded that the necessity of placing a stent remains unclear.

Wang et al. analyzed 22 randomized controlled trials and concluded that, despite the mentioned tubeless advantages, ureteral stents were valuable in preventing hospitalizations [8]. 

Reicherz et al. evaluated the use of short-term UC stenting in pre-stented patients. They concluded that patients can be treated with short-term ureteral stenting or tubeless procedures after uncomplicated URS [4]. 

Post-operative pain control is of vital importance, particularly in the current opiate crisis era.

Kang et al. evaluated opiate exposure after ureteroscopy. Among 208 patients, 12% were prescribed opiates in the first 30 days after the surgery and 7% continued the prescriptions [16]. 

Berger et al. found stent insertion to be associated with an increased risk of persistent as well as chronic use of opiates [17]. 

We found that the post-operative VAS score was significantly lower in the tubeless subgroup with most patients reporting no pain at all, and in agreement with previous studies, we did not find a difference between the stented and UC subgroups in the post-operative VAS scores [5,7,13,15].

Furthermore, most of the tubeless patients who required analgesics were treated with Dipyrone and did not require morphine. The UC group patients had significantly more pain and discomfort, requiring significantly larger amounts of Tramadol and morphine.

There are several limitations to this study, most notably it being a retrospective study. Our database is collected prospectively, but its retrospective analysis enabled us to objectively produce three relevant groups and analyze drainage type effect and pain control in the postoperative course.

## 5. Conclusions

In this study, it is shown that post-operative drainage by UC or DJS is still common practice. Drainage with UC is a remnant of the pre-high frequency laser and dusting era, supported by a concern of large obstructing fragments, and should be avoided in simple ureterorenoscopies due to significant patient discomfort with no evidence of benefit. The tubeless approach is feasible with less post-operative pain and complications.

## 6. Brief Summary

Post-ureteroscopy tubeless drainage is feasible in selected cases with less postoperative pain and complications in comparison to UC and stent drainage.

## Figures and Tables

**Table 1 jpm-12-01878-t001:** Patients and stone characteristics.

n = 269		UC	Stent	Tubeless	*p*-Value
Number of patients		136/269(50%)	63/269(24%)	70/269(26%)	
Max diameter of biggest stone(mm)		8.4(6.1–12)	12(8.6–16.6)	8(5.2–11.5)	<0.001
Total stone volume(mm^3^)		201.4(85.8–419)	550(237.6–1251.4)	140.6(65.5–336.3)	<0.001
HU(mean)		864(512–1194)	1031(737–1356)	727(500–1330)	0.002
Stone location	Ureter	44/136(32%)	13/63(20%)	17/70(24%)	0.119
	Kidney	68/136(50%)	45/63(71%)	40/70(57%)	
	Both	12/136(9%)	2/63(3%)	4/70(6%)	
	Other	11/136(8%)	3/63(5%)	9/70(13%)	
Pre-drained	Stent	56/136(41%)	5/63(8%)	41/70(59%)	<0.001
	PCN	3/136(2%)	5/63(8%)	1/70(>1%)	0.066
Scope used	Rigid	42/136(31%)	9/63(14%)	17/70(24%)	0.027
	flex	59/136(43%)	42/63(67%)	41/70(59%)	
	Both	33/136(24%)	12/63(19%)	12/70(17%)	
Op.Time(min)		32(23–45)	49(33–60)	28(20–40)	<0.001
Hospitalization		1.94 ± 1.2	1.8 ± 0.5	1.2 ± 2.2	0.56
ER Visit 1		27/136(20%)	10/63(16%)	2/70(3%)	<0.001
ER Visit 2		3/136(2%)	2/62(3%)	0	0.365
Re-Hospitalization		16/136(12%)	6/63(10%)	1(>1%)	0.04
Complications overall		36/136(26%)	11/63(17%)	7/70(10%)	
Complications(Clavien-Dindo)	I–II	33/136(24%)	8/63(13%)	7(10%)	0.002
	IlIa	1	0	0	
	IlIb	2(1%)	3/63(5%)	1	
	IV	0	0	0	
Complications	Renal colic/SS	27/136(20%)	4/63(6%)	7/70(10%)	
	ARF	5/136(4%)	1	1	
	Sepsis/Fever	13/136(10%)	7/63(11%)	0	
Follow up (days)		67 ± 32.1	77 ± 32.5	56.4 ± 27.5	
Lost to long FU		22/136(16%)	7/63(11%)	9/70(13%)	
Auxillary proc.	1	19/136(14%)	32/63(51%)	1	<0.001
	2	4/136(3%)	5/63(8%)	0	
Stone free overall		102/124(82%)	54/56(96%)	58/61(95%)	0.285

**Table 2 jpm-12-01878-t002:** Postoperative VAS groups.

	(DJS)	(UC)	(T)
VAS 0	29/61(48%)	44/103(43%)	33/49(67%)
VAS 0-4	3/61(5%)	15/103(15%)	20/49(41%)
VAS 5-7	23/61(38%)	40/103(39%)	14/49(29%)
VAS 8-10	5/61(8%)	9/103(9%)	1/49(2%)
Average VAS	2.82 ± 2.56	2.87 ± 2.78	1.75 ± 2.79

**Table 3 jpm-12-01878-t003:** Postoperative analgesic use.

	(DJS)	(UC)	(T)	*p*
Dipyron (gr/case)	32/61 = 0.5	75/103 = 0.7	11/49 = 0.22	0.002
Tramadol (mg/case)	2700/61 = 44.3	4700/103 = 45.6	1300/49 = 26.5	0.034
Diclofenac (mg/case)	150/61 = 2.5	450/103 = 4.4	0	0.047
MO (mg/case)	10/61 = 0.2	60/103 = 0.6	0	0.354

## Data Availability

Not applicable.

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
