# Peer review of "Tubeless Ureterorenoscopy-Our Experience Using a 120 W Laser and Dusting Technique: Postoperative Pain, Complications, and Readmissions"

_jpm, 2022, doi:10.3390/jpm12111878_

Round 1

Reviewer 1 Report

The authors conducted a retrospective study assessing the differential perioperative outcomes between three drainage methods after URS with lithotripsy. However, a Cochrane review and meta-analysis (ref #1), including only RCTs, showed that stenting was associated with worse pain on POD4 or thereafter. Still, there were no differences in the other endpoints evaluated. Therefore, this study does not yield any novelty over the known evidence. In addition, in this study, these three groups are highly affected by selection bias; instead, they select the drainage methods based on their stone/patient characteristics and/or perioperative findings. This kind of comparative study must always be a “fair comparison” between groups. Thus, results from this study did not add findings to this field and affect our clinical practice.

In addition, a three-arm comparison is sometimes confusing and methodologically complex. Methodologically, the authors should perform a Bonferroni analysis to detect which two groups have differences.

To improve this manuscript, I would suggest some comments focused on abstract.

・English should be revised by a native speaker.

・Line23: What is T? I think this is tubeless.

・Line 28-33: The authors just explain the selection bias. I believe that the primary message from this study is that the tubeless approach was safely performed without any postoperative UTI and/or ureteral stricture (how about?). These results are more important rather than differential patient demographics (selection bias)

・Line 37-38: Where this statement comes from? The authors never assessed the efficacy of powerful laser dusting in this study.

・Finally, a ureter catheter (UC) is often confused with a urethral catheter! Therefore, I prefer ureteral stenting (US) drainage.

Author Response

Dear Reviewer,

First and foremost we would like to thank you for the time you dedicated to revise our manuscript. It is GREATLY appreciated. We hope, that following our response you will find our paper suitable for acceptance to the Journal of Personalized Medicine.

The authors conducted a retrospective study assessing the differential perioperative outcomes between three drainage methods after URS with lithotripsy. However, a Cochrane review and meta-analysis (ref #1), including only RCTs, showed that stenting was associated with worse pain on POD4 or thereafter. Still, there were no differences in the other endpoints evaluated. Therefore, this study does not yield any novelty over the known evidence. In addition, in this study, these three groups are highly affected by selection bias; instead, they select the drainage methods based on their stone/patient characteristics and/or perioperative findings. This kind of comparative study must always be a “fair comparison” between groups. Thus, results from this study did not add findings to this field and affect our clinical practice.

In addition, a three-arm comparison is sometimes confusing and methodologically complex. Methodologically, the authors should perform a Bonferroni analysis to detect which two groups have differences.

To improve this manuscript, I would suggest some comments focused on abstract.

・English should be revised by a native speaker.

・Line23: What is “T”? I think this is tubeless.

・Line 28-33: The authors just explain the selection bias. I believe that the primary message from this study is that the tubeless approach was safely performed without any postoperative UTI and/or ureteral stricture (how about?). These results are more important rather than differential patient demographics (selection bias)

・Line 37-38: Where this statement comes from? The authors never assessed the efficacy of powerful laser dusting in this study.

・Finally, a ureter catheter (UC) is often confused with a urethral catheter! Therefore, I prefer ureteral stenting (US) drainage.

  • We agree with many of the mentioned comments. This is indeed a retrospective cohort with limitations that accompany the model. The aim of this study was to reinforce the relatively new concept of tubeless ureteroscopy, to add additional aspects of post-operative pain including opiate use and to evaluate the commonly used ureter catheter as a drainage option. We believe that our finding reinforce endourology practice to use the tubeless option, in the appropriately selected cases, and to omit UC as a post operative drainage option.
  • All of the other comments were addressed and corrected in the text and we highly appreciate them.

English was revised and edited.

T was replaced with tubeless

The efficacy of powerful laser was removed

Reviewer 2 Report

I congratulate the authors for the study called Tubeless ureterorenoscopy. Our experience using 120-W laser and dusting technique. Postoperative pain, complications and readmissions.

The study presents interesting concept of suggestion to increase the tubeless for URS procedure, however some parts are unclear.

1)Please correct Tubless with Tubeless in line 71 and 115

2)Stortz with Storz line 78, is the scope the reusable flex X2 ? 

How have you divided the 2 groups? How did you decide who did not need a stent and who did? Please clarify if this is a retrospective study or prospective.

No mention was made about the laser settings used 

I believe that the patients with UC stayed overnight and had the UC removed in day 1. Did the stent also stay overnight? How can you explain the same length of stay between the 2 groups? The UC groups has higher attendance to the emergency department. How do you explain this considering that they get discharged without UC? why do they go back to the emergency and require more painkillers ?

Was ureteric access sheath used in any of the groups? this can affect the need to of DJS placement. This needs to be specified in the text.

Please clarify the nature of the IIIb complications. Why does the UC groups have such high type I-II complications? If they had the UC removed after24 hours , what is the main issue with them?

Author Response

Dear Reviewer,

First and foremost we would like to thank you for the time you dedicated to revise our manuscript. It is GREATLY appreciated. We hope, that following our response you will find our paper suitable for acceptance to the Journal of Personalized Medicine.

The study presents interesting concept of suggestion to increase the tubeless for URS procedure, however some parts are unclear.

1)Please correct Tubless with Tubeless in line 71 and 115

Corrected

2)Stortz with Storz line 78, is the scope the reusable flex X2 ? 

Yes, a flexible ureteroscope flex X2. Corrected and incorporated in the text

How have you divided the 2 groups? How did you decide who did not need a stent and who did? Please clarify if this is a retrospective study or prospective.

Bigger stones, the use of ureteric access sheath and relatively more complicated cases were drained by DJS, which we usually leave for up to two weeks. In some cases which required complementary procedures, a DJS was left as well. In uneventful cases, two surgeons left UC as their routine and the third completed the procedures as tubeless.

No mention was made about the laser settings used:

Thank you for your comment. The laser lithotripsy was done using Holmium:Yag laser (PulseP 120 H, Lumenis Inc., Israel). The following laser settings were used (dusting in situ 0.6-0.8 Joule,12-15 Hz; fragmentation 0.8-1 Joule, 8-10 Hz; pop dusting 0.6 Joule 50 Hz).  Laser settings were incorporated into the text

I believe that the patients with UC stayed overnight and had the UC removed in day 1. Did the stent also stay overnight? How can you explain the same length of stay between the 2 groups? The UC groups has higher attendance to the emergency department. How do you explain this considering that they get discharged without UC? why do they go back to the emergency and require more painkillers ?

Thank you for the comment! All three groups stayed for 1 night as a part of hospital routine. Why the UC group had higher rates of post-operative emergency room visits is in our opinion a key question for this study. The UC group is a heterogeneous group. In retrospect, it can be presumed that patients who did not require pain killers and had no post operative ER visits could have been left tubeless, while the rest should have had a DJS. We believe that according to this data UC should be omitted.

Was ureteric access sheath used in any of the groups? this can affect the need to of DJS placement. This needs to be specified in the text.

Thank you for the comment, this information was added to the text.

Page 2, sentence 77

Please clarify the nature of the IIIb complications. Why does the UC groups have such high type I-II complications? If they had the UC removed after24 hours , what is the main issue with them?

We had two cases of patients drained with UC who needed DJS insertion due to fever and Stone Street.

Round 2

Reviewer 1 Report

The authors addressed the issues the reviewer raised.

However, the quality is not satisfactory. The authors should have revised and responded point-by-point.

In addition, this reviewer does not agree with the authors’ methodology and consecutive conclusions at all. The authors should reconsider the study design, focusing point of the study, and interpretation of results.

As previously mentioned, this study never proved postoperative safety due to significant selection bias that easier cases were selected for tubeless management; patients with tubeless are more likely to have small, less total stone volume, short operation time, and hospitalization, I guess younger and healthy patients (data has not shown). The authors only can mention that postoperative pain levels and analgesic use were significantly lower in the tubeless group, with a significant reduction in opiate usage. But in the Abstract, this statement accounts for only a limited part of the results.

In total, this paper is not scientific. If the authors just wanted to present their real-world practice data, it’s OK. But, when the authors state the results, the limitation of selection bias must be emphasized everywhere in the manuscript. The authors only can say that “In the real-world practice of our institution, tubeless management following ureterorenoscopy were likely to be selected in easy case (small stone or short operation time); therefore, this selection bias makes the true differential postoperative complication rates obscure. However, postoperative pain levels and analgesic use were significantly lower in the tubeless group. Tubeless management seems to be feasible in improving the patient QOL in well-selected patients. “

If the authors want to compare the complications and readmission rates, there were several methodological flaws.

Basically, we should consider matching the patient characteristics using the propensity score to reduce selection bias between the two different treatment procedures (specifically in the current topic). Therefore, I would recommend the two-arm comparison (for example, UC vs. tubeless because DJS is only used for complex cases or in case of complementary procedures required).

When the authors stick to a three-arms comparison (despite unfair comparison), as mentioned in the first revision, the authors should perform a Bonferroni analysis to detect which two groups have significant differences in all analyses.

Finally, the authors also stick to the dusting procedure with a 120-W laser; this term is also in the Title. However, if the surgeon used only a semi-rigid scope (around 20% of entire cohorts), fragmentation was performed, and used N-circle to extract the stone fragment. If the authors want to include “dusting” in the Title, only patients treated with a flexible scope should be included. Or the Title should be changed.

Author Response

Dear Reviewer,

First and foremost we would like to thank you again for the time you dedicated to revise our manuscript. It is GREATLY appreciated. We hope, that following our response you will find our paper suitable for acceptance to the Journal of Personalized Medicine.

The scope of this paper was to present retrospective data on postoperative results of three drainage options and to emphasize the feasibility of tubeless ureteroscopies with postoperative results and poor results of UC drained patients (same stone characteristics as in tubeless cases).

We completely agree that there is a bias in case selection and we defiantly believe that a prospective randomized trial is warranted to reconfirm our findings.  The dusting procedure with a 120-W laser reduced the concern of large obstructing fragments and significantly reduced the need for the evaluated UC drainage.

We are planning a prospective trial to reevaluate our results, but for now we hope you find our finding interesting to your readers and suitable for publication.

Thank you very much!

Reviewer 2 Report

Thanks for making some of the requested changes, however some more clarification is required 

Was ureteric access sheath used in any of the groups? this can affect the need to of DJS placement. This needs to be specified in the text but was not specified. Thank you for the comment, this information was added to the text.

Page 2, sentence 77   - This is a very vague reply

How many cases needed UAS? How many of the UAS had DJ ? Did the use of UAS influence the decision for DJ placement?

Please add nature of IIIb complications in the results.

Also please specify the complications of UC group in the results. What are the 23% of I-II complications ?

Author Response

Dear Reviewer,

First and foremost we would like to thank you again for the time you dedicated to revise our manuscript. It is GREATLY appreciated. We hope, that following our response you will find our paper suitable for acceptance to the Journal of Personalized Medicine.

Thanks for making some of the requested changes, however some more clarification is required 

Was ureteric access sheath used in any of the groups? this can affect the need to of DJS placement. This needs to be specified in the text but was not specified. Thank you for the comment, this information was added to the text.

Page 2, sentence 77   - This is a very vague reply –

We apologize for the vague answer.

We used ureteric access sheath in cases with big complexed stones, and as you mentioned most of those cases did require DJS placement. We incorporated this data into methods sections.  

How many cases needed UAS? How many of the UAS had DJ ? Did the use of UAS influence the decision for DJ placement?

We used UAS in 7 cases. All this cases were drained with DJS. The decision was defiantly influenced by UAS because of the potential ureteric edema/preventing stricture.

Please add nature of IIIb complications in the results.

Also please specify the complications of UC group in the results. What are the 23% of I-II complications ?

Thank you for the comment. The complications were incorporated into the text.

CD IIIb complications were two cases of patients drained with UC that needed DJS insertion due to fever and Stone Street. CD I-II complications included Atelectasis and fever requiring antipyretics and electrolyte abnormalities.